# A Randomised Crossover Trial of Behaviour Guidance Techniques on Children with Special Health Care Needs during Dental Treatment: The Physiological Variations

**DOI:** 10.3390/children9101526

**Published:** 2022-10-05

**Authors:** Norsaima Ismail, Khairil Anuar Md Isa, Ilham Wan Mokhtar

**Affiliations:** 1Centre for Paediatric Dentistry & Orthodontics Studies, Jalan Hospital, Faculty of Dentistry, Universiti Teknologi MARA, Sungai Buloh Campus, Sungai Buloh 47000, Selangor, Malaysia; 2Department of Basic Sciences, Faculty of Health Science, Universiti Teknologi MARA, Puncak Alam Campus, Bandar Puncak Alam 42300, Selangor, Malaysia; 3Centre for Comprehensive Care Studies, Jalan Hospital, Faculty of Dentistry, Universiti Teknologi MARA, Sungai Buloh Campus, Sungai Buloh 47000, Selangor, Malaysia

**Keywords:** passive immobilisation, physiological response, Papoose board, children with special health care needs

## Abstract

Passive immobilisation is regarded as able to potentially cause physical distress and intense anxiety manifestations. The study aims to investigate the physiological variations of children with special health care needs while using a Papoose board and a combination of basic behaviour guidance during dental treatment. This is a randomised crossover trial involving 90 children with special health care needs receiving standard dental care with two methods of behaviour guidance sequentially. Exposure A is a combination of tell-show-do, distraction, and positive reinforcement, while exposure B is passive immobilisation with a Papoose board. The subject child’s blood pressure, heart rate, and oxygen saturation level were measured at four different times during dental treatment. In total, 74 children’s physiological data were successfully collected with a mean age of 9.85 years (SD = 2.71). Further, 64.9% of the children were diagnosed with autism spectrum disorder, 12.2% with attention deficit hyperactivity disorder, 9.5% with intellectual disability, 8.1% with Down syndrome, 2.2% with global developmental delay, and 1.1% with dyslexia and cerebral palsy, respectively. The measurement of children’s blood pressure, heart rate, and oxygen saturation level with the application of a Papoose board or a combination of the basic behaviour guidance revealed no significant changes (*p* > 0.05). The use of a Papoose board is safe and has no discernible influence on the child’s physiological responses.

## 1. Introduction

Behaviour guidance for children with special health care needs (CSHCN) can be challenging due to dental anxiety, communication problems, and limited understanding of dental treatment, which may cause them to manifest resistant behaviours [1]. These children frequently exhibit high anxiety levels and display uncooperative behaviours, which form barriers to regular dental treatment in a conventional dental environment [2]. Although the management for the dental treatment itself is similar to that of healthy children, they might require different behaviour guidance strategies in dentistry. Protective stabilisation is a method to restrict a patient’s movement, with or without the patient’s permission, to reduce the risk of injury while allowing safe completion of treatment [3]. If the restriction involves another person(s), it is considered active immobilization, while on the other hand, passive immobilisation techniques utilise the use of mechanical restraining devices. The use of full-body immobilisation devices such as Papoose board in paediatric dentistry, however, remains controversial [4]. Earlier studies revealed that the experience of immobilization devices would be a traumatic event for children and had a significant role in the development of dental anxiety, thus potentially evoking a negative attitude toward future dental treatment [5,6]. They viewed the Papoose board as a stressor that can evoke dental anxiety and elevates internal stress in children who experienced restraining device [7]. On the contrary, a study showed that a Papoose board can be used as a Sensory Adaptation Technique (SAT) device that provides deep touch pressure to stabilise and tranquilise patients with special health care needs during anxiety-provoking situations [8]. This calming tool significantly assists medical or dental staff in managing uncooperative or anxious children and minimises the difficulties and risks during interventions [9]. Theoretical articles have suggested that the homeostasis of the autonomic nervous system (ANS) plays a role in anxiety and stress [10]. It can physiologically manifest as increased heart rate and blood pressure that represent sympathetic activation parameters of autonomic measures (catecholamine action) [11]. Blood pressure (BP) and heart rate (HR) are variables that can conveniently be measured to reflectively quantify the physiological response to anxiety and interpreted as physiological measures of stress during dental treatment [12,13]. Although studies showed that detailed and positive explanations improved parental acceptance toward the use of a Papoose board for children during dental treatment [14,15,16]; the utilisation of the so-called “aversive technique” remained imposed with difficult ethical evaluations when the dentists make individual assessments [17]. The study aims to investigate the physiological variations of heart rate, blood pressure, and oxygen saturation level for children with special health care needs while using a Papoose board, and a combination of tell-show-do, distraction, and positive reinforcement techniques during dental treatment. The outcomes are expected to provide insight into physiological evidence toward this controversial technique leading to improved efficiency and safety of dental treatment for children with special health care needs.

## 2. Materials and Methods

### 2.1. Study Design

This is a longitudinal randomised crossover trial that incorporates two visits in two months intervals. In total, 90 children with special health care needs who registered for dental treatment at Special Care Dentistry (Paediatric) Clinic that fulfilled inclusion and exclusion criteria were recruited with convenience sampling. Simple randomization with the application of sequentially numbered, opaque, sealed envelopes (SNOSE) method of allocation concealment is adopted. The sample size was calculated using G*Power 3.1 Software with effect size (r): 0.25, type I error (α): 0.05, and power (1-β err prob): 0.8 at 4 measurements of time point on a single group which yielded a minimum of 24 samples. The same sample cohort was involved in earlier survey research on their parental perceived mannerisms towards the use of passive immobilisation during dental treatment for their CSHCN. Hence the minimum number of the total sample was calculated determined at 74 samples. The ethics were approved by the University X Research Ethics Committee with the reference number (REC/08/2020/FB 189) and registered in the clinical trial registry of International Standard Randomized Control Trial Number (ISRCTN) (ISRCTN57204958). The study was conducted in accordance with the Declaration of Helsinki and written consent was obtained.

### 2.2. Study Population

#### 2.2.1. Inclusion Criteria

The study population consisted of children with special health care needs aged 16 years old and below who attended the Special Care Dentistry (Paediatric) Clinic of Faculty of Dentistry X from 1 October 2020 to 1 October 2021. They were either diagnosed with neurodevelopmental disorder and/or physical disabilities and willing to sit on the dental chair with minimal physical persuasion either by the parent/caregivers and/or the operating team. The children were able to be treated with either prophylaxis treatment using a brush bur or simple composite restorative treatment (ICDAS 03) without local anaesthesia administration and rubber dam placement using small round bur with a slow-speed rotary handpiece. The subject child also must never have any previous exposure to the Papoose board experience.

#### 2.2.2. Exclusion Criteria

Medically compromised children with any kind of underlying heart or lung disease that may affect cardiovascular measurements were excluded from the trial protocol. On the day of the intervention, the children must not suffer from acute pain and be free from any influence of medication.

### 2.3. Study Visits and Procedures

Ninety caregivers of children with special health care needs who presented at the centre were given subject information sheets explaining what participation entailed by the principal investigator. Written consent was obtained prior to the envelope selection. The study was a longitudinal design in which subjects received two treatments (exposures) sequentially over two periods. The order in which treatments were received was randomised following the AB/BA exposure sequences. Exposure A was a combination of basic behaviour guidance, which consists of tell-show-do (TSD), distraction (D), or positive reinforcement (PR), while exposure B was passive immobilisation with a Papoose Board^®^ (Figure 1). All children were approached with standard communication-based behaviour protocol throughout the research activities.

#### Workflow for AB/BA Sequencing

During the first visit, standard oral examination and treatment plans were performed and proposed. The subject child was intended to receive dental treatment as planned; either dental prophylaxis or simple restorative treatment without local anaesthesia administration and rubber dam placement using a slow-speed rotary handpiece. Physiological changes measurements were recorded by a designated research assistant. The subject child’s blood pressure (BP), heart rate (HR), and oxygen saturation level (SPO_2_) were measured by using Welch Allyn Connex^®^ Vital Signs Monitor (HillromTM, Chicago, IL, USA) at four different phases (Table 1).

The respondents were placed in a washout period for 1–2 months to diminish the impact of the carryover effect before proceeding to the second visit for the subsequent Exposure B. Subject child repeated the same procedure as in the first visit of workflow. The exposure sequence will be vice versa for B-A sequence. Figure 2 shows the research flowchart.

### 2.4. Statistical Analysis

Data entry and analysis were performed using the IBM SPSS software (Version 28.0, IBM Knowledge Center, USA). A value of *p* < 0.05 is considered statistically significant. Repeated measures ANOVA were used to compare four physiological means, which were systolic blood pressure (SBP), diastolic blood pressure (DBP), heart rate (HR), and oxygen saturation level (SPO_2_) within groups at four different times. This analysis was to assess any significant difference in physiological impacts on baseline phase, pre-treatment phase, treatment phase, and post-treatment phase in CSHCN using a Papoose board (PB) and a combination of basic behaviour guidance technique (BGT) during dental treatment.

## 3. Results

### 3.1. Descriptive Analysis

Eighty-eight children completed the sequence treatment and two patients failed to attend the second visit. Only seventy-four children’s physiological data can be successfully collected during the study protocol. Fourteen children with incomplete physiological data were excluded for physiological variance analysis. The mean age of the children was 9.85 years (SD = 2.71) with most of the children (79.7%) aged 6–11 years followed by early adolescents (16.2%) and the least (4.1%) aged between 2–5 years old. The male patients were double the sample (70.3%) compared to the female (29.7%). Among all, about 64.9% of the children were diagnosed with autism spectrum disorder, followed by attention deficit hyperactive disorder, 12.2%; 9.5% were intellectual disability, 8.1% were children with Down syndrome, 2.7% had global developmental delay, and the rest were dyslexia (1.1%) and cerebral palsy (1.1%), respectively. When utilizing the Papoose board, sixty CSHCN received prophylactic full mouth polishing treatment using a bristle brush and fourteen children underwent ICDAS 03 composite restorative treatment. As for sequence treatment, 40 children were able to complete (B-A) sequence in contrast with (A-B) sequence completed by 34 children. On average, all the patients fulfilled both study visits in 2.92 months (SD = 1.13).

### 3.2. Multivariate Analysis

A one-way repeated measured analysis of variance (ANOVA) was performed to investigate the physiological changes in systolic blood pressure (SBP), diastolic blood pressure (DBP), heart rate (HR), and oxygen saturation level (SPO_2_) for CSHCN while using the Papoose board and a combination of tell-show-do, distraction, and positive reinforcement technique during dental treatment (Table 2).

#### 3.2.1. Systolic Blood Pressure (SBP)

ANOVA analysis indicated a non-significant time effect, Wilks’ Lambda = 0.99, F (3144) = 0.34, *p* = 0.80, η^2^ = 0.07. This reveals that there is no change in children’s SBP when measured during baseline, preoperatively, intraoperatively, and postoperatively with the application of two types of BGT during dental treatment. A follow-up comparison indicated that each pairwise difference was also not significant (*p* = 0.80). Estimated marginal means of patients’ SBP while using passive immobilisation technique displayed increasing patterns from baseline to pre-operative time (intervention of PB) followed by a slight decrease during intra-operative (procedural time) and continued to reduce even after removal of the devices (post-operative). On the contrary, by applying a combination of basic BGT, the children showed a decreasing pattern in mean SBP from about 3.82 mmHg until intra-operative time and increased back to nearly the same level as the original baseline mean (Figure 3a).

#### 3.2.2. Diastolic Blood Pressure (DBP)

The mean DBP in CSHCN while using PB showed a similar increasing pattern with mean SBP from baseline to pre-operative of about 7.32 mmHg and decreased significantly from pre-operative to intra-operative time and continued to drop down to post-operative (Figure 3b). While applying a combination of basic BGT, children’s mean DBP steadily declined from baseline (78.15 mmHg) to intra-operative (75.45 mmHg) and suddenly went up to the level with the baseline value. Although, the mean values were not dramatically changed for both interventions; however, the mean values while using PB were faintly higher during pre-operative and intra-operative as compared to the combination of basic BGT. The results of the repeated measures ANOVA indicated a non-significant time effect with Wilks’ Lambda= 0.98, F (3144) = 1.17, *p* = 0.32, η^2^ = 0.24. In a nutshell, there were no changes in children’s DBP measured during baseline, pre-operatively, intra-operatively, and post-operatively with the application of two types of BGT during dental treatment (*p* = 0.32).

#### 3.2.3. Heart Rate (HR)

The cumulative mean value of HR remained constant at 88 beat/min before and during intervention with the PB, but later started to climb up during the procedural time and gently fell into baseline value. Conversely, the combination of basic BGT exhibit constant escalation until it reaches the same intra-operative mean values as PB intervention and later fell back to the baseline value. Pre-operative mean HR while using a combination of basic BGT was higher (90 beat/min) compared to PB (88 beat/min) (Figure 4). Conclusively, the usage of either PB or a combination of basic BGT during dental treatment in CSHCN did not produce any statistically significant changes in children’s HR (*p* = 0.13). The results of ANOVA analysis indicated a non-significant time effect of heart rate measured during baseline, pre-operatively, intra-operatively, and post-operatively with the application of two types of BGT during dental treatment.

#### 3.2.4. Oxygen Saturation Level (SPO_2_)

Graph for cumulative mean values of oxygen saturation level (SPO_2_) at four different times for CSHCN receiving two types of BGT during dental treatment showed a zig-zagged pattern (Figure 5). The ANOVA analysis revealed a statistically non-significant time effect, Wilks’ Lambda = 0.99, F (3144) = 0.42, *p* = 0.74, η^2^ = 0.01. This showed that applying either PB or a combination of basic BGT during dental treatment does not cause any increase in children’s oxygen saturation levels.

## 4. Discussion

Out of 88 children who completed the sequence, 14 were excluded from physiological data analysis as they were unable to tolerate the blood pressure cuff tightness, excessive movement, and disruption during the physiological measurement (10 children from A-B sequence and 5 from B-A sequence). They were restless and refused to stay in the dental chair for further treatment. Each child was then assigned to a waiting list for a procedure under general anaesthesia to continue the treatment. One approach to managing withdrawal was by per-protocol (PP) analysis. The PP analysis excludes children who did not meet the eligibility criteria after randomisation and those who were noncompliant with the treatment protocol. The advantage of PP analysis is its simplicity. Nevertheless, the application of PP analysis was merely for a trial that intended to analyse the impact of an intervention on the participant who complied with the trial protocols [18]. Even with the loss of 17% sample dropout (*n* = 14), the result is still within 0.8 statistical power and can be considered valid. Total blinding of the participants and the operator to the usage of behaviour methods to the outcome was impractical. Hence to minimise the possible bias, the operator was blinded to the outcomes of the physiological measures as the person responsible for data entry was performed by a designated research assistant.

Patient with autism spectrum disorder (ASD) is the highest among our samples and boys are twice the number compared to girls. This is well reported by Centres for Disease Control (CDC) in 2018 where most of the children diagnosed with autism spectrum disorder (ASD) are boys, four times more likely than girls [19]. The imbalance between different groups of disabilities among all the samples is due to the convenience sampling method that was used during the data collection process, which is likely to cause inherent selection bias [20]. In addition, refined inclusion criteria, which require the patient to be cardiovascular fit without taking any medication, restrict the subject selection. As compared to ASD, the current study managed to include only six children with Down’s syndrome as the prevalence of cardiovascular anomalies in Down’s syndrome was reported to be as high as 42% [21].

The sympathetic and parasympathetic branches of the autonomic nervous system (ANS) have excitatory and inhibitory effects on heart function, respectively. In stressful events, the sympathetic system is activated and generally elevates heart rate and cardiac contractility [22]. The study revealed that there were no significant differences in the vital sign readings among CSHCN while receiving dental treatment using a Papoose board or a combination of tell-show-do, distraction, and positive reinforcement techniques. All children who received both groups of interventions maintained stability and within the normal range of vital signs values throughout the dental procedures. The outcomes expressed that applying a Papoose board during dental treatment does not cause abnormal variation, either increasing or decreasing in children’s vital signs. The study showed contradicting results from a study performed in Japan where uncooperative healthy children (3–5 years old) showed a relatively high heart rate under restraint while receiving restorative and root canal treatment [7]. The distinct methodological background and studied population differences may yield different outcomes.

Mean SBP and DBP showed an increasing pattern as CSHCN started to wear the Papoose board during the pre-operative time; however, it slowly dropped when we started with the suggested dental procedures. This could be due to initial hesitation during the application of the devices, either the blood pressure cuff set, the Papoose board, or both. Subsequently, when the treatment started, they became calmer and more receptive hence reducing their blood pressure. Although the reduction in blood pressure during the procedure was not statistically significant, there was evidence that Papoose board application as a sensory adaptation technique can trigger the effect of deep touch pressure in stressful dental situations [8]. According to Marshall et al., 20% of children with autism experienced a calming effect related to a stabilization device [23].

The study discovered that children treated with a combination of TSD, PR, and D had slightly higher mean HR than children treated with Papoose board during the pre-operative time, indicating that sensory hypersensitivity may serve a significant role in the development of dental anxiety in children with autism spectrum disorder [24]. According to our observation, when the operator displayed the rotary brush on the child’s nail and proximity using the tell-show-do technique, it could increase a child with autism’s hyperresponsiveness to sensory input. Intraoperative mean HR was identical for both interventions (90.97 beats/min). This suggests that regardless of the type of BGT employed, CSHCN’s HR will increase during dental procedures. This is explained by the delay in the psychosocial, cognitive, and emotional development of individuals with special needs, who may exhibit a higher level of anxiety [25]. They may have difficulty appreciating the invasive aspect of dental treatment or comprehending the dentist’s instructions and explanations, thereby increasing their level of tension [26].

Our investigation observed no significant changes in the child’s SPO_2_ throughout Papoose board application, as it fluctuated within a 1% increment or reduction. This showed that Papoose board is considered safe when used properly [27]; however, children with respiratory diseases such as asthma, cystic fibrosis, complications of obesity, restriction of the chest, or full supine positioning may require extra caution. Inappropriate placement of a stabilisation device around the chest may impair the children’s respiration [5]. Thus, continuous monitoring of the children’s physical well-being and tightness of the stabilisation device was mandatory throughout the procedure, which included respiratory and circulatory status, skin integrity, and vital signs [28]. Advanced training is advised prior to the implementation of protective stabilisation on patients in order to provide safe dental treatment [28,29].

The existing study highlighted the concern related to safety when utilizing passive immobilisation devices in a dental setting. Despite being dismissed as inappropriate and banned by law in certain continents, such as Northern Europe and Australia [30], the study findings challenge the ethical principles of nonmaleficence, autonomy, and justice when it is used based on the principle of the beneficence of restraint in paediatric dentistry [17]. Various beliefs and attitudes on the usage of passive immobilisation devices involve numerous contributing factors which may lead to either positive or negative circumstances [15,16,31]. The outcome observed and recognised that the use of passive immobilisation is not for convenience but is considered a safe alternative method in providing dental treatment for children with special health care needs.

The study has the following strengths as it was a randomised crossover clinical trial, in which the risk of confounding factors can be minimized because a comparison of two behaviour guidance interventions was measured on the same participant and the subject child serves as their own control. Moreover, the treatment and technique of applying the Papoose board were performed by a single operator that has substantial training in dentistry. The study was conducted in a controlled environment to reduce the influence of the external environment.

## 5. Conclusions

Our research established that when a Papoose board is used properly, it is considered safe and has no discernible influence on the child’s physiological variations in the studied population. This study offered physiological evidence that the full-body passive immobilisation technique can be considered reliable and applicable in paediatric dentistry, particularly among children with special health care needs. We proposed that the utilisation of vital sign monitoring is advisable in order to detect any significant physiological changes during restraint. In selecting the behaviour guidance techniques, it is advisable to observe individual factors of the receiver (such as cognitive and behaviour).

It is suggested that future studies can focus on each specific type of disability group as we foresee that different types of disabilities have distinct and specific characteristics which might give a different result.

## Figures and Tables

**Figure 1 children-09-01526-f001:**
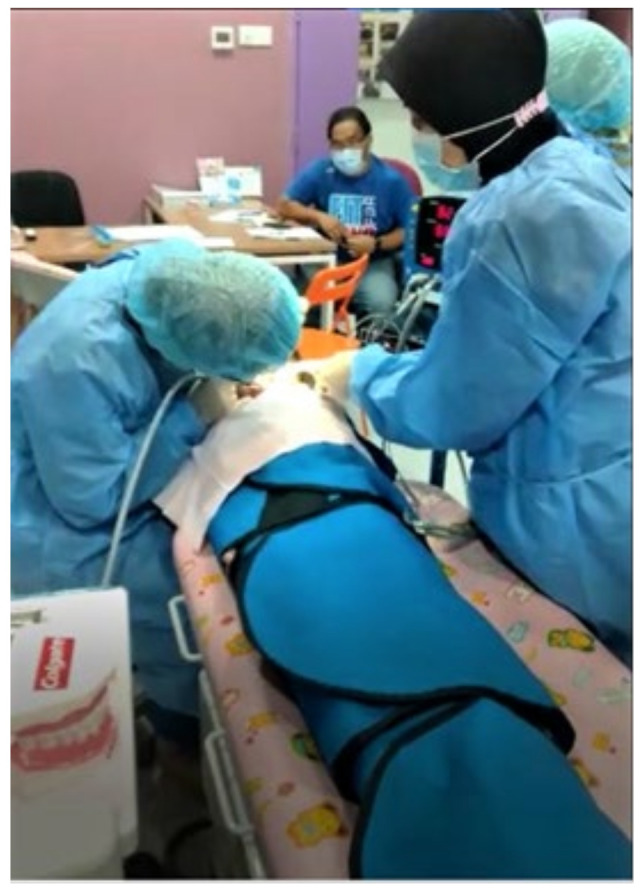
The subject child was wrapped with Olympic Papoose Board^®^ (Olympic Papoose Board, Natus Medical Incorporated, Pleasanton, CA, USA) during dental treatment.

**Figure 2 children-09-01526-f002:**
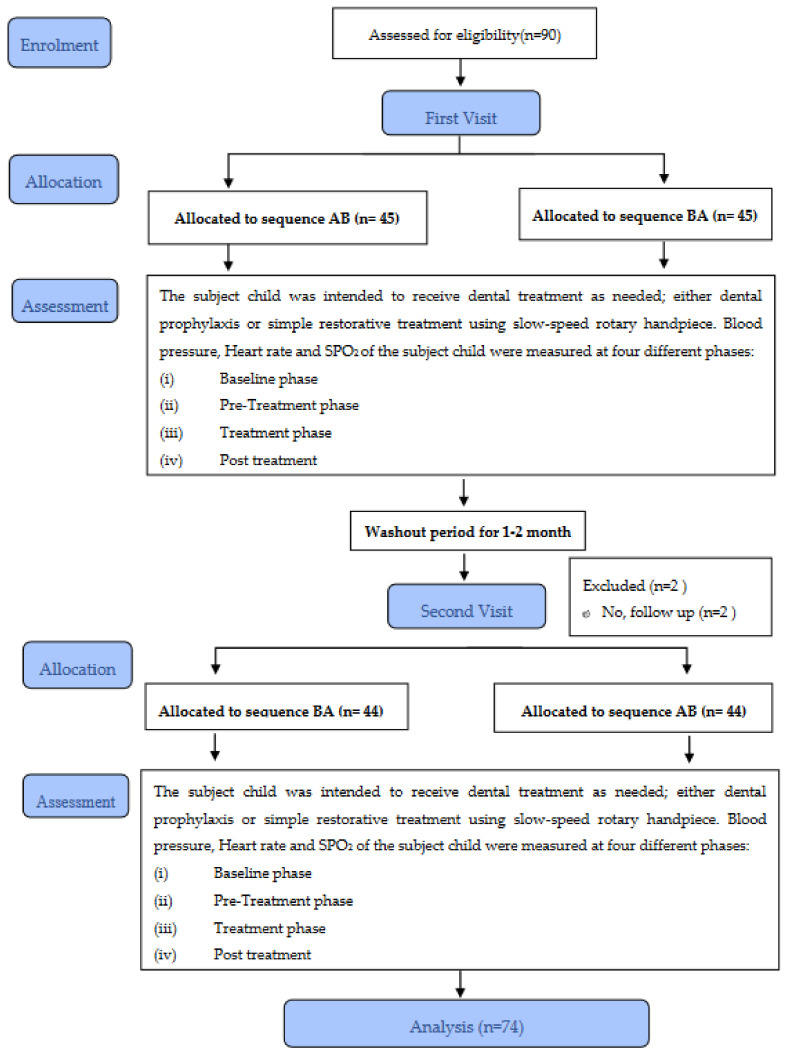
A CONSORT flow diagram for crossover trials.

**Figure 3 children-09-01526-f003:**
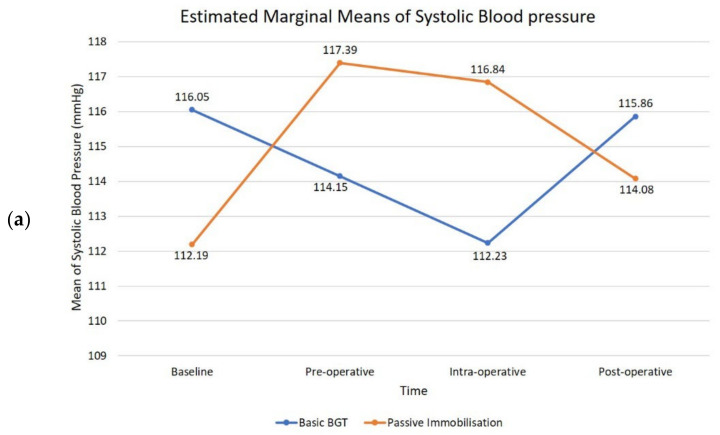
Graph pattern of cumulative means values of (**a**) systolic blood pressure; (**b**) diastolic blood pressure at four time periods.

**Figure 4 children-09-01526-f004:**
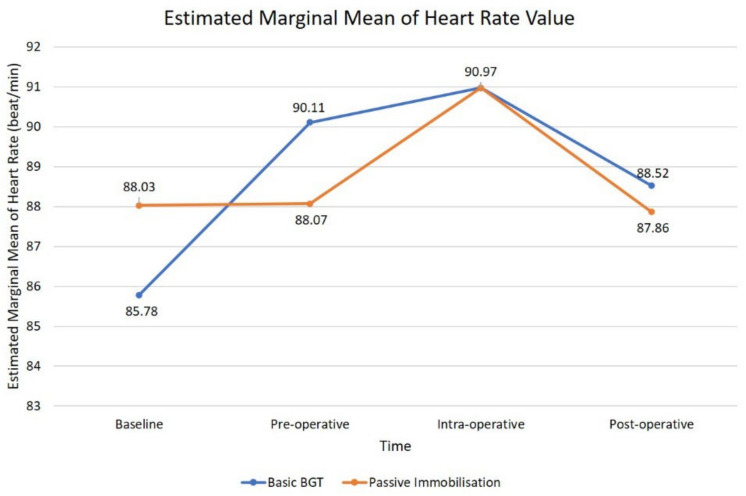
Graph pattern of cumulative means values of heart rate at four time periods.

**Figure 5 children-09-01526-f005:**
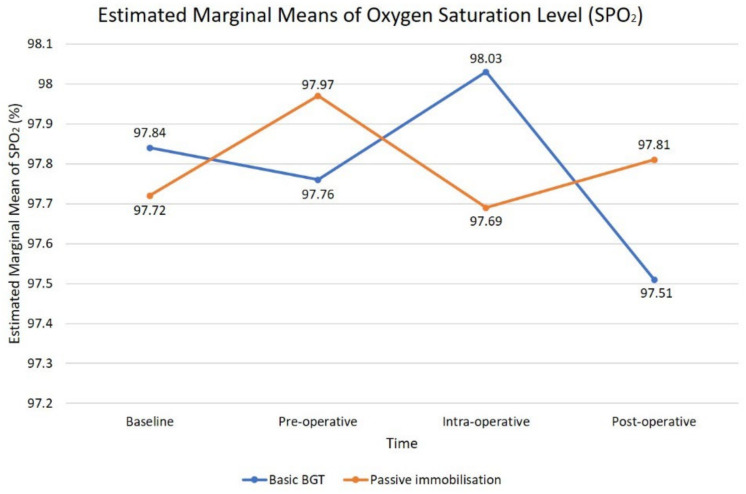
Graph pattern of cumulative means values of oxygen saturation level at four time periods.

**Table 1 children-09-01526-t001:** Phases of time measurement of physiological vital signs.

Phase	Time of Measurement
Baseline	Subject child in the waiting room/surgery room sitting on chair
Pre-treatment	Subject child lying down on the dental chair and proposed BGT (Exposure A) or wrap in Papoose board (Exposure B) will be applied to subject child for 1–2 min.
Treatment	Chosen treatment either prophylaxis or caries excavation (using slow speed handpiece; inside the oral cavity for 1–2 min.
Post-treatment	Subject child remains lying on the dental chair without Exposure A or Exposure B which lasted 1–2 min.

**Table 2 children-09-01526-t002:** Multivariate analyses of repeated measures (ANOVA).

	MEAN (SD)		F TEST (DF)	PARTIAL ETA SQUARED (η^2^)			*p*-VALUE	
	PAPOOSE BOARD	BASIC BGT						
**SYSTOLIC BLOOD PRESSURE (mmHg)**			0.34 (3144) *	0.01			0.80	
Baseline	112.19 (18.95)	116.05 (21.27)						
Pre-operative	117.39 (23.50)	114.15 (18.95)						
Intra-operative	116.84 (21.14)	112.23 (19.92)						
Post-Operative	114.08 (19.14)	115.86 (19.42)						
**DIASTOLIC BLOOD** **PRESSURE (mmHg)**			1.17 (3144) *	0.02			0.32	
Baseline	73.96 (15.94)	78.15 (17.69)						
Pre-operative	81.28 (25.21)	77.22 (16.64)						
Intra-operative	77.66 (18.273)	75.45 (17.68)						
Post-operative	75.15 (15.56)	77.99 (20.06)						
**HEART RATE (beat/min)**			1.93 (3144) *	0.04			0.13	
Baseline	88.03 (14.51)	85.78 (18.47)						
Pre-operative	88.07 (19.64)	90.11 (18.09)						
Intra-operative	90.97 (20.50)	90.97 (20.50)						
Post-operative	87.86 (17.20)	88.52 (17.89)						
**OXYGEN SATURATION LEVEL (%)**			0.42 (3144) *	0.01			0.74	
Baseline	97.72 (2.02)	97.84 (2.20)						
Pre-operative	97.97 (2.13)	97.76 (2.35)						
Intra-operative	97.69 (4.49)	98.03 (1.86)						
Post-operative	97.81 (1.67)	97.51 (3.84)						

* Wilks’ Lambda Test.

## Data Availability

Not applicable.

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
