# Peer review of "A Randomised Crossover Trial of Behaviour Guidance Techniques on Children with Special Health Care Needs during Dental Treatment: The Physiological Variations"

_children, 2022, doi:10.3390/children9101526_

Round 1
Reviewer 1 Report
Abstract: First sentence--passive immobilization does not always cause intense anxiety manifestations and physical distress; I would suggest stating "potentially cause."
Introduction, line 60: I would delete "few" as there are more references to support this claim than any other made in the introduction.
Line 63: I would not use the word "inflicted"
M&M line 76: I would state "convenience sampling." Line 78: use past tense. Line 79: I would suggest stating what specific effect size for what variable was used in the calculation and state the results of the power analysis.
"Simple restorative treatment" should be further defined (local anesthesia, isolation).
Line 98: use "influence"
Were children in the papoose board treated without any verbal explanation at all? Typically communication-based behavior guidance is used with a restrained child as done otherwise. Was the child simply restrained and not introduced to dental procedures with TSD or thanked for cooperation with PR? It seems unkind to not talk to a child, especially when restrained.
Was pre-enrollment behavior assessed (i.e. with the Frankl scale)? Were cooperative subjects restrained?
Results: why were physiological data not successfully collected on all subjects--was it because of physical movement/disruption? This would imply a higher level of stress in this group. Related to this question is the fact fewer A-B subjects completed--is it related to the restraint? The discussion states that 14 subjects were unable to tolerate the BP cuff--was this the only reason and seen in both groups? Further data on these subjects is necessary.
Were any procedures aborted because of concerns with safety in either group? Were there any data collected or assessments made regarding intraoperative behavior--FLACC scale, Frankl scale, etc.?
Discussion: line 233: "2:1 to girls" is unclear. Line 237: convenience vs. convenient. Line 254: I would appreciate more discussion about why these results vary from citation 7. What was different about these subjects?
Conclusion: The conclusions should be tailored to the population studied, especially if there is any indication on the level of cooperation of the subjects prior to the study procedures.
Reviewer 2 Report
This is a well-written research article that has been produced following a randomised crossover trial of children with special needs during dental treatment. This is a randomised crossover trial involved 90 children with special health care who received standard dental care with two methods of behaviour guidance sequentially.
One of those methods of behaviour guidance was passive immobilization with papoose board. This is a contentious issue for dental practitioners, parents and children alike. Instead of being dismissed as inappropriate in 2022, this article is important because it investigates the physiological variations of children with special health care needs while using a papoose board. Some countries do allow (and it is considered as common practise) for the physical restraint of children.
However, as some countries (including Australia) do not allow any form of immobilisation of children in any setting, this fact needs to be made explicit. Any form of immobilisation would be construed as physical abuse.
